# Reproducibility Study of "ITI-GEN: Inclusive Text-to-Image Generation"

**Daniel Gallo Fernández**
*University of Amsterdam*

**Răzvan-Andrei Matișan**
*University of Amsterdam*

**Alejandro Monroy Muñoz**
*University of Amsterdam*

**Janusz Partyka**
*University of Amsterdam*

**Reviewed on OpenReview:** *https://openreview.net/forum?id=d3Vj360Wi2*

## Abstract

Text-to-image generative models often present issues regarding fairness with respect to certain sensitive attributes, such as gender or skin tone. This study aims to reproduce the results presented in "ITI-GEN: Inclusive Text-to-Image Generation" by Zhang et al. (2023a), which introduces a model to improve inclusiveness in these kinds of models. We show that most of the claims made by the authors about ITI-GEN hold: it improves the diversity and quality of generated images, it is scalable to different domains, it has plug-and-play capabilities, and it is efficient from a computational point of view. However, ITI-GEN sometimes uses undesired attributes as proxy features and it is unable to disentangle some pairs of (correlated) attributes such as gender and baldness. In addition, when the number of considered attributes increases, the training time grows exponentially and ITI-GEN struggles to generate inclusive images for all elements in the joint distribution. To solve these issues, we propose using Hard Prompt Search with negative prompting, a method that does not require training and that handles negation better than vanilla Hard Prompt Search. Nonetheless, Hard Prompt Search (with or without negative prompting) cannot be used for continuous attributes that are hard to express in natural language, an area where ITI-GEN excels as it is guided by images during training. Finally, we propose combining ITI-GEN and Hard Prompt Search with negative prompting.

## 1 Introduction

Generative AI models that solve text-to-image tasks pose a series of societal risks related to fairness. Some of them come from training data biases, where certain categories are unevenly distributed. As a consequence, the model may ignore some of these categories when it generates images, which leads to societal biases on minority groups.

In order to tackle this issue, Zhang et al. (2023a) introduce Inclusive Text-to-Image Generation (ITI-GEN), a method that generates inclusive tokens that can be appended to the text prompts. By concatenating these fair tokens to the text prompts, they are able to generate diverse images with respect to a predefined set of attributes (e.g. gender, race, age). For example, we can add a "woman" token to the text prompt "a headshot of a person" to ensure that the person in the generated image is a woman.

In this work, we aim to focus on:

- **[Reproducibility study] Reproducing the results from the original paper.** We verify the claims illustrated in Section 2 by reproducing some of the experiments of Zhang et al. (2023a).

- **[Extended Work] Proxy features.** Motivated by attributes for which ITI-GEN does not perform well, we carry out experiments to study the influence of diversity and entanglement in the reference image datasets.

- **[Extended Work] Generating images using negative prompts.** Hard Prompt Search (HPS) (Ding et al., 2021) with Stable Diffusion (Rombach et al., 2022) is used as a baseline in the paper. We study the effect of adding negative prompts to Stable Diffusion as an alternative way of handling negations in natural language and compare the results with ITI-GEN. We also highlight the potential of combining negative prompting with ITI-GEN, using each method for different types of attributes.

- **[Extended Work] Modifications to the original code.** We improve the performance at inference by fixing a bug that prevented the use of large batch sizes, integrate ITI-GEN with ControlNet (Zhang et al., 2023b), as well as provide the code to run our proposed method that handles negations. At the same time, we include bash scripts to make it easy to reproduce our experiments.

In the next section, we introduce the main claims made in the original paper. Then, we describe the methodology of our study, highlighting the models and datasets that we used, as well as the experimental setup and computational requirements. An analysis of the results and a discussion about them will follow.

## 2 Scope of reproducibility

In the original paper, the authors make the following **main claims**:

1) **Inclusive and high-quality generation.** ITI-GEN improves inclusiveness while preserving image quality using a small number of reference images during training. The authors support this claim by using KL divergence and FID score (Heusel et al., 2017) as metrics.

2) **Scalability to different domains.** ITI-GEN can learn inclusive prompts in different scenarios such as human faces and landscapes.

3) **Plug-and-play capabilities.** Trained fair tokens can be used with other similar text prompts in a plug-and-play manner. Also, the tokens can be used in other text-to-image generative models such as ControlNet (Zhang et al., 2023b).

4) **Data and computational efficiency.** Only a few dozen images per category are required, and training and inference last only a couple of minutes.

5) **Scalability to multiple attributes.** ITI-GEN obtains great results when used with different attributes at the same time.

In this work, we run experiments to check the authors' statements above. Additionally, we study some failure cases of ITI-GEN and propose other methods that handle negations in natural language.

## 3 Methodology

The authors provide an open-source implementation on GitHub[1] that we have used as the starting point. To make our experiments completely reproducible, we design a series of bash scripts for launching them. Finally, since the authors did not provide any code for integrating ITI-GEN with ControlNet (Zhang et al., 2023b), we implement it ourselves to check the compatibility of ITI-GEN with other generative models. All of these are better detailed in Section 3.4.

---

[1] `https://github.com/humansensinglab/ITI-GEN`

### 3.1 Model description

ITI-GEN is a method that improves inclusiveness in text-to-image generation. It outputs a set of fair tokens that are appended to a text prompt in order to guide the model to generate images in a certain way. It achieves this by using reference images for each attribute category. For example, if we use the prompt "a headshot of a person" and provide images of men and women, the model will learn two tokens, one for each gender.

Given a set of $M$ attributes where each attribute $m$ has $K_m$ different attribute categories, ITI-GEN learns a set of fair tokens that represent each attribute category. For each combination, the corresponding fair tokens are aggregated and concatenated to the original text prompt $\boldsymbol{T}$ to build a inclusive prompt $\boldsymbol{P}$. We denote the set of all inclusive prompts with $\mathcal{P}$, and the set of all inclusive prompts that correspond to category $i$ of attribute $m$ with $\mathcal{P}_i^m$.

The training procedure involves two losses. The first one is the directional alignment loss, which relates the provided reference images to the inclusive prompts. In order to compare images and text, CLIP (Radford et al., 2021) is used to map them to a common embedding space using its text encoder $E_{\text{text}}$ and image encoder $E_{\text{img}}$. Following the original code, the ViT-L/14 pre-trained model is used. The directional alignment loss is defined by

$$\mathcal{L}_{\text{dir}} = \sum_{m=1}^{M} \sum_{1 \leq i < j \leq K_m} 1 - \langle \Delta_{\boldsymbol{I}}^m(i,j), \Delta_{\boldsymbol{P}}^m(i,j) \rangle, \tag{1}$$

where $\Delta_{\boldsymbol{I}}^m(i,j)$ is the difference of the average of image embeddings between categories $i$ and $j$ of attribute $m$, and $\Delta_{\boldsymbol{P}}^m(i,j)$ is the difference between the average of embeddings of inclusive prompts correspondent to those categories.

This loss is replaced by a cosine similarity loss $\mathcal{L}_{\text{cos}}$ when it is undefined (i.e. when the batches are not diverse enough and do not contain samples for every attribute category). Given an image of a certain attribute category, we want to make it similar to all inclusive prompts that contain that attribute category.

The second loss is the semantic consistency loss, which acts as a regularizer by making the original text prompts similar to the inclusive prompts:

$$\mathcal{L}_{\text{sem}} = \sum_{m=1}^{M} \sum_{1 \leq i < j \leq K_m} \max_{\boldsymbol{P} \in \mathcal{P}_i^m \cup \mathcal{P}_j^m} \left(0, \lambda - \langle E_{\text{text}}(\boldsymbol{P}), E_{\text{text}}(\boldsymbol{T}) \rangle \right), \tag{2}$$

where $\boldsymbol{T}$ is the raw text prompt and $\boldsymbol{P}$ the inclusive text prompt for an attribute category combination including category $i$ or $j$ for attribute $m$ and $\lambda$ is a hyperparameter.

In this way, the total loss for a batch is defined by:

$$\mathcal{L}_{\text{total}} = \begin{cases} \mathcal{L}_{\text{dir}} + \mathcal{L}_{\text{sem}} & \text{if } \mathcal{L}_{\text{dir}} \text{ is defined,} \\ \mathcal{L}_{\text{cos}} + \mathcal{L}_{\text{sem}} & \text{otherwise.} \end{cases} \tag{3}$$

We invite the reader to check the original paper (Zhang et al., 2023a) for a more detailed explanation of the model.

At inference, the output of the generative model will reflect the attribute categories of the fair tokens. In this way, sampling over a uniform distribution over all combinations leads to fair generation with respect to the attributes of interest. In addition, the text prompt can be the same that was used for learning (in-domain generation) or different (train-once-for-all).

The generative model must be compatible with the encoder (CLIP in this case). Following the original paper, we use Stable Diffusion v1.4 (Rombach et al., 2022) for most of the experiments. We also show compatibility with models using additional conditions like ControlNet (Zhang et al., 2023b) in a plug-and-play manner.

An alternative to ITI-GEN is HPS (Ding et al., 2021), which works by specifying the categories directly in the original prompt. For example, we could consider the prompt "a headshot of a woman" to generate a woman. One problem with this approach is that it does not handle negation properly and using a prompt like "a headshot of a person without glasses" will actually increase the chances of getting someone with glasses. However, this can be circumvented by using the *unconditional_conditioning* parameter, which is hard-coded to take in the empty string in the current Stable Diffusion sampler (PLMS). By providing the features that are not wanted we can prevent them from appearing in the generated images. For example, we can pass "eyeglasses" to get a headshot of someone without them.

A denoising step in Stable Diffusion takes an image $\boldsymbol{x}$, a timestep $t$ and a conditioning vector $\boldsymbol{c}$. This outputs another image that we will denote by $f(\boldsymbol{x}, t, \boldsymbol{c})$. The technique involves two conditioning vectors, $\boldsymbol{c}$ and $\bar{\boldsymbol{c}}$ (the negative prompt), and outputs

$$\lambda(f(\boldsymbol{x}, t, \boldsymbol{c}) - f(\boldsymbol{x}, t, \bar{\boldsymbol{c}})) + f(\boldsymbol{x}, t, \bar{\boldsymbol{c}}),$$

where $\lambda$ is the scale, that we set to the default 7.5 as we explain later in Section 3.3. We call this Hard Prompt Search with negative prompting and we will refer to it as HPSn throughout the paper.

## 3.2 Datasets

The authors provide four datasets[2] of reference images to train the model:

- **CelebA** (Liu et al., 2015), a manually-labeled face dataset with 40 binary attributes. For each attribute, there are 200 positive and negative samples (400 in total).

- **FAIR** (Feng et al., 2022), a synthetic face dataset classified into six skin tone levels. There are 703 almost equally distributed images among the six categories.

- **FairFace** (Karkkainen & Joo, 2021), a face dataset that contains annotations for age (9 intervals), gender (male or female) and race (7 categories). For every attribute, there are 200 images per category.

- **Landscapes HQ (LHQ)** (Skorokhodov et al., 2021), a dataset of natural scene images annotated using the tool provided in Wang et al. (2023). There are 11 different attributes, each of them divided into five ordinal categories with 400 samples per category.

## 3.3 Hyperparameters

In order to reproduce the results of the original paper as closely as possible, we use the default training hyperparameters from the code provided. In a similar way, for image generation, we use the default hyperparameters from the ControlNet (Zhang et al., 2023b) and Stable Diffusion (Rombach et al., 2022) repositories (for HPS and HPSn). Moreover, we generate images with a batch size of 8, which is the largest power of two that can fit in an NVIDIA A100 with 40 GB of VRAM.

## 3.4 Experimental setup and code

The code used to replicate and extend the experiments can be found in our GitHub repository[3]. We made two minor changes to the authors' implementation: adding a seed in the training loop of ITI-GEN to make it reproducible and fix a bug in the script for image generation to handle batch sizes larger than 1. We also provide bash scripts to replicate all our experiments easily.

Moreover, the authors do not provide any code for combining ITI-GEN with ControlNet (Zhang et al., 2023b). Thus, we implement it ourselves on top of the pre-existing ControlNet code[4] in order to validate the plug-and-play capabilities of ITI-GEN with other text-to-image generation methods. More specifically,

---

[2]https://drive.google.com/drive/folders/1_vwgrcSq6DKm5FegICwQ9MwCA63SkRcr
[3]https://github.com/amonroym99/iti-gen-reproducibility
[4]https://github.com/lllyasviel/ControlNet

Table 1: **Inclusiveness with respect to single attributes.** KL divergences between the obtained and the uniform distributions over the attribute category combinations. Reference images are from CelebA. To classify the images, CLIP and manual labeling are used. The text prompt is "a headshot of a person". An extended version of these results can be found in Appendix A.

| Method | | Male | Young | Pale Skin | Eyeglasses | Mustache | Smiling |
|---|---|---|---|---|---|---|---|
| HPS | CLIP | 0.000000 | 0.000000 | 0.000416 | 0.387506 | 0.158797 | 0.000046 |
| | Reported | 0.000010 | 0.027000 | 0.002800 | 0.371000 | 0.241000 | 0.004400 |
| ITI-GEN | CLIP | 0.000000 | 0.026869 | 0.001156 | 0.015053 | 0.280577 | 0.000000 |
| | Manually labeled | 0.000000 | 0.001156 | 0.000185 | 0.000000 | 0.000000 | 0.000000 |
| | Reported | 0.000002 | 0.000200 | 0.000000 | 0.000200 | 0.000450 | 0.002500 |

we use the *conditioning* parameter of the DDIM sampler (which is meant to be initialized with the text prompt in the original implementation) to inject ITI-GEN's inclusive prompt. At the same time, we used the *unconditional_conditioning* parameter of the sampler to guide the model not to generate, for example, black-and-white or sepia images. We run experiments using depth, canny edge, and human pose as additional conditions for ControlNet. For more information about how to use the integration of ITI-GEN with ControlNet, we invite the reader to check our GitHub repository[5].

For reproducing the experiments using HPS (Ding et al., 2021), we use Stable Diffusion's code[6]. We modify the text-to-image generation script to receive an optional negative prompt for HPSn.

We consider two aspects when we evaluate the results: fairness and image quality. To measure fairness, we first generate images for every category combination. Then we classify all images using CLIP (Cho et al., 2023; Chuang et al., 2023). As explained in the original paper, this turned out to be unreliable (even when not using negation), so we manually labeled the images for some attributes. Finally, we compare the obtained distribution with a uniform distribution with respect to all attribute categories of interest by calculating the Kullback–Leibler (KL) divergence. For quantifying the image quality, we computed the Fréchet Inception Distance (FID) score (Heusel et al., 2017; Parmar et al., 2022), which compares the distribution of generated images with the distribution of a set of real images. The score depends on the number of images involved so, for reproducibility reasons, we used the number of generated images mentioned by the authors (i.e. approximately 5,000). When we compare two methods (e.g. ITI-GEN and HPS), we also perform human evaluations by displaying images from the same attribute category combination for each method side by side, making the human subject choose the one with higher quality. We used 714 images for HPS, HPSn and ITI-GEN, respectively.

### 3.5 Computational requirements

We perform all experiments on an NVIDIA A100 GPU. Training a single attribute for 30 epochs takes around a minute for CelebA, 3 minutes for LHQ, 4 minutes for FAIR and less than 5 minutes for FairFace. For the four datasets, we use 200 images per category (or all of them if there are less than 200). Generating a batch of 8 images takes around 21 seconds (less than 3 seconds per image). It is also possible to run inference on an Apple M2 chip, although it takes more than 30 seconds per image. In total, our reproducibility study adds up to at most 20 GPU hours.

All our experiments were run using the Snellius infrastructure located in the Amsterdam Data Tower. The data center reports a Power Usage Effectiveness (PUE) of 1.19 [7]. Thus, using the Machine Learning Emissions Calculator (Lacoste et al., 2019), we obtain that our experiments emitted around 6 kg of $CO_2$.

---

[5]`https://github.com/amonroym99/iti-gen-reproducibility`
[6]`https://github.com/CompVis/stable-diffusion`
[7]`https://www.clouvider.com/amsterdam-data-tower-datacentre/`

Table 2: **Image generation quality.** FID scores for CelebA dataset (lower is better). As reference dataset, we use the Flickr-Face-HQ Dataset (FFHQ) (Karras et al., 2019). We set different seeds to generate images, which might explain why we get a better score (the authors do not report their generation method).

| Method | Number of images | FID score | |
|---|---|---|---|
| | | Ours | Reported |
| Vanilla Stable Diffusion | 5,040 | 78.34 | 67.40 |
| HPS | 5,040 | 70.55 | – |
| HPS (with negative prompting) | 5,040 | 65.08 | – |
| ITI-GEN (30 epochs) | 5,040 | 54.86 | 60.38 |

# 4 Results

Our reproducibility study reveals that the first four claims mentioned in Section 2 are correct, whereas the last one does not hold when the number of attributes increases. In this section, we first highlight the results reproduced to support or deny the main claims of the authors. Then, we compare HPSn with ITI-GEN for the cases when ITI-GEN struggles to generate accurate images.

## 4.1 Results reproducing original paper

### 4.1.1 Inclusive and high-quality generation

To verify the first claim, we train ITI-GEN on every attribute of the CelebA dataset and generate 104 images (13 batches of size 8) per category. Considering there are 40 binary attributes, that makes a total of $40 \times 2 \times 104 = 8,320$ images.

For every attribute we have $104 \times 2 = 208$ images that are classified using CLIP. Since this classification is not very reliable, we also label the images of selected attributes manually. After that, we compute the KL divergence. In Table 1 we see how HPS struggles with attributes that require negation (e.g., "a headshot of a person with no eyeglasses"), while ITI-GEN performs well.

Since the FID score reported in the paper uses around 5,000 images, and we have 8,320, we decide to compute the FID score using the last 63 images of every category, which results in a total of 5,040 images. The results are in Table 2, where we can see that ITI-GEN obtains the best score.

We also carry out human evaluations to compare the quality of the images generated by HPS and HPSn with ITI-GEN's. The results show that, in 52.24 % of the cases, humans prefer images generated with HPS over ITI-GEN. The same thing happens with HPSn in 56.58% of the cases. This shows a discrepancy with the interpretation of the FID scores, where ITI-GEN shows slightly better quality than the HPS methods. A possible explanation is that the reference datasets for the computation of the FID score and for training ITI-GEN (FFHQ and CelebA respectively) might be similar, which would imply a bias towards ITI-GEN.

### 4.1.2 Scalability to different domains

Figure 1 shows that it is possible to apply ITI-GEN to multiple domains, as the second claim holds. For the human faces generated with the "Skin tone" attribute, we are able to compute the FID score in the same way as before (using the FFHQ dataset), and obtain 46.177. For the natural scenes generated with the "Colorful" attribute, we compute the FID score by comparing the generated images with the training ones, obtaining 62.987. Note that both figures are within a similar range as the ones in Table 2.

### 4.1.3 Plug-and-play capabilities

We perform a number of experiments to verify the third claim, demonstrating in the end that it holds. More specifically, we show that inclusive tokens learned with one prompt can be applied to other (similar) prompts without retraining the model, as it is shown in Figure 2. We train the binary "Age" and "Gender"

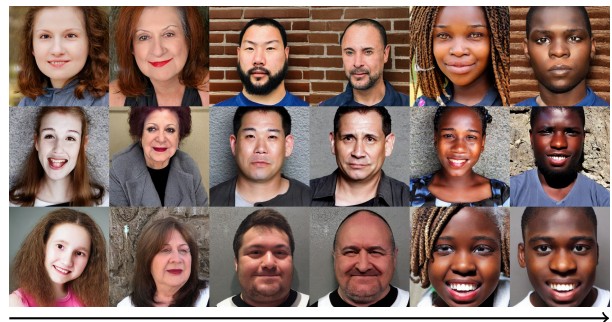
(a) "Skin tone" applied to "a headshot of a person".

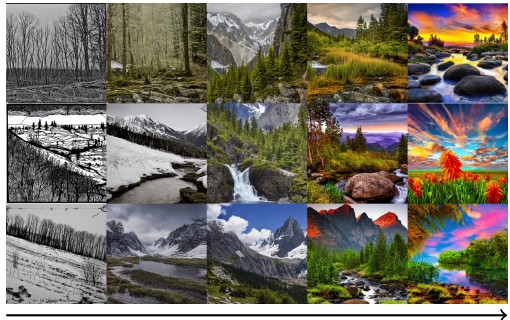
(b) "Colorful" applied to "a natural scene".

Figure 1: **Scalability to different domains.** Images generated with ITI-GEN in two different domains: human faces and natural scenes. Each column corresponds to a different category of the attribute.

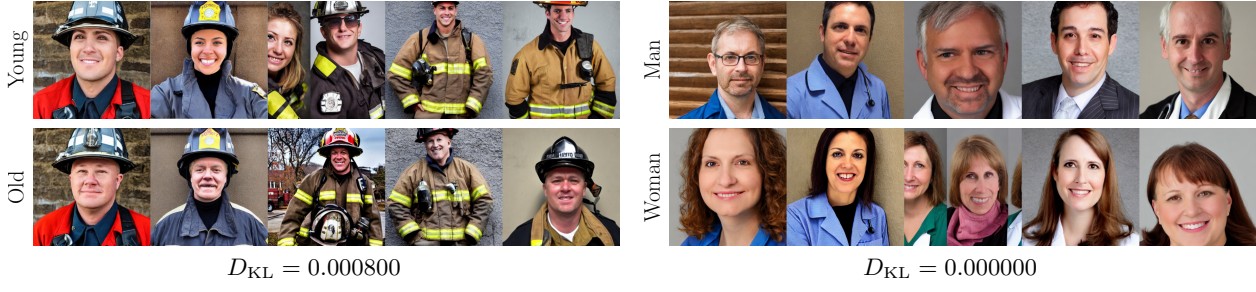

$D_{\mathrm{KL}} = 0.000800$

$D_{\mathrm{KL}} = 0.000000$

(a) "Age" token applied to "a headshot of a firefighter".   (b) "Gender" token applied to "a headshot of a doctor".

Figure 2: **Plug-and-play capabilities.** ITI-GEN is used to learn inclusive tokens for "Age" and "Gender" using the text prompt "a headshot of a person". These tokens are then applied to other similar text prompts.

tokens with the prompt "a headshot of a person" using CelebA dataset and apply them to "a headshot of a firefighter" and "a headshot of a doctor" prompts. However, while the generated images are still diverse, their quality diminished. Thus, we obtain slightly higher FID scores, i.e., 152.94 and 157.6 for the firefighter and doctor examples, respectively.

In addition, we run experiments to illustrate the compatibility of ITI-GEN with ControlNet (Zhang et al., 2023b) in a plug-and-play manner. Figure 3 depicts the desired behaviour: ITI-GEN improves diversity of ControlNet with respect to the inclusive tokens used. For this experiment, we train the "Age" token with "a headshot of a person" prompt using FairFace dataset, which has 9 categories for this attribute, and apply it to "a headshot of a famous woman" prompt.

However, we can observe that the newly introduced fair tokens may entangle other biases that are inherent from the reference images such as hair color, clothing style or skin tone. This is an observation the original authors also make but they did not try to find an explanation for it stating that the "disentanglement of attributes is not the primary concern of this study". Therefore, we do believe ITI-GEN improves diversity of ControlNet, but only with respect to the attributes of interest, even though it may bring some other biases taken from the reference images. For additional results, please refer to Appendix B.

### 4.1.4   Data and computational efficiency

In order to verify the claim about data efficiency, we generate images for the attributes "Colorful", "Age", and "Skin tone" using different number of reference images per category. As in the original paper, we find ITI-GEN to be robust in the low-data regime. The reader can refer to Appendix D for more information. Regarding computational efficiency, training takes less than 5 minutes and generation takes around 2 seconds per image on our hardware, as we show in Section 3.5.

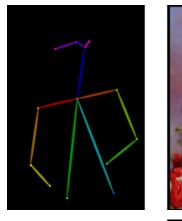 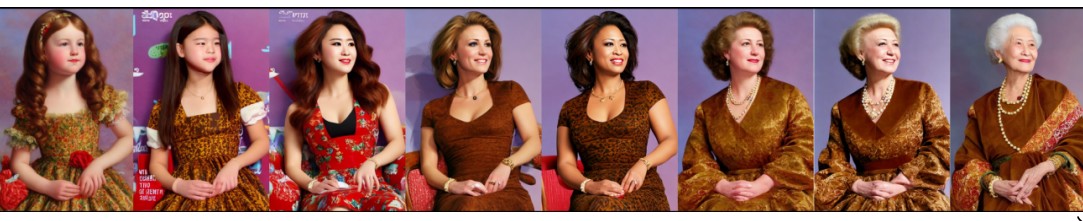

Age

Figure 3: **Compatibility with ControlNet.** We generate images using the prompt "photo of a famous woman" and human pose (left) as additional condition. The attribute of interest is "Age", which is trained using the text prompt "a headshot of a person".

### 4.1.5 Scalability to multiple attributes

Table 3 illustrates how ITI-GEN performs for some combinations of binary attributes. We can observe that while the KL divergence is low when we use two attributes, it increases significantly when we keep adding attributes. Thus, this claim is not entirely correct, as we further investigate this observation qualitatively and quantitatively in the next section, where we also highlight more of ITI-GEN's pitfalls.

Table 3: **Inclusiveness with respect to multiple attributes.** Comparison of the KL divergence between the obtained and the uniform distributions. All reference images are from CelebA dataset. The text prompt is "a headshot of a person". We use CLIP to classify images. We also include the results for HPS with negative prompting, which we further discuss in Section 4.2.

| Method | | Male × Young | Male × Young × Eyeglass | Male × Young × Eyeglass × Smile |
|---|---|---|---|---|
| HPS | CLIP | 0.000990 | 0.125403 | 0.032695 |
| | Authors | 0.003500 | 0.399000 | 0.476000 |
| HPSn | CLIP | 0.000990 | 0.000475 | 0.000273 |
| ITI-GEN | CLIP | 0.051310 | 0.378709 | 0.356023 |
| | Authors | 0.000130 | 0.061000 | 0.094000 |

## 4.2 Results beyond original paper

### 4.2.1 Proxy features

ITI-GEN might use some undesired features as a proxy while learning the inclusive prompts for certain attributes. This is the case for attributes like "Bald", "Mustache" and "Beard", which seem to use "Gender" as proxy. One could argue that this is not important as long as the model is able to generate people with and without the desired attribute. However, it is actually a problem because as we show in Figure 4, some pairs of attributes are strongly coupled and ITI-GEN is not able to generate accurate images for all elements in the joint distribution. HPSn, on the other hand, is able to disentangle the attributes. More examples are shown in Appendix C.

Our main hypothesis is that the reason for this lies within the reference images. If we inspect them, we see that most of the bald people are men, which may be the reason why the model is using gender as a proxy. To delve into this, we perform an experiment using two variants of the "Eyeglasses" dataset (see Table 4), which is diverse and works fine when combined with the "Gender" attribute. Then, we test the ability of the model to learn the joint distribution using those reference images.

Figure 5a illustrates with quantitative and qualitative results how ITI-GEN correctly handles the combination of the "Gender" and "Eyeglasses" attributes using the original reference dataset. If we alter the diversity of the dataset by entangling both attributes, the model fails to learn the "Eyeglasses" attribute and uses "Gender" as a proxy feature, as shown in Figure 5b (the generated samples of women with eyeglasses are

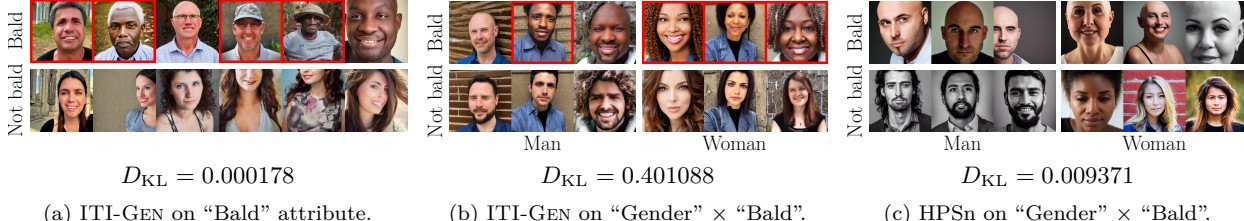

$D_{KL} = 0.000178$

(a) ITI-Gen on "Bald" attribute.

$D_{KL} = 0.401088$

(b) ITI-Gen on "Gender" × "Bald".

$D_{KL} = 0.009371$

(c) HPSn on "Gender" × "Bald".

Figure 4: **Proxy features are used by the model for certain attributes. (a)** Generated samples using the fair tokens for "Bald". All positive samples are men, whereas all negative samples are women, which indicates that "Gender" might be used as a proxy feature. **(b)** When combining the "Gender" and "Bald" attributes, ITI-Gen fails to generate samples of bald women. **(c)** HPS with negative prompting is able to accurately generate bald women. In (b) and (c), the KL divergence is computed using 104 manually labeled samples.

Table 4: **Non-diverse "Eyeglasses" reference datasets. (a)** is the original dataset form CelebA. In **(b)**, the negative samples are only of women, and, the positive ones, of men. In **(c)**, only men are included.

|  | Negative | Positive |
|---|---|---|
| (a) Original | Men without eyeglasses
Women without eyeglasses | Men with eyeglasses
Women with eyeglasses |
| (b) Gender-biased | Women without eyeglasses | Men with eyeglasses |
| (c) Male-only | Men without eyeglasses | Men with eyeglasses |

just men). However, if only men are included in the reference images, the method does learn the feature correctly (Figure 5c), because the only difference between the negative and positive samples are the glasses, and not the gender. This comparison demonstrates that we must be careful choosing the reference images, as the model might not learn what we expect, but the most salient difference between the datasets.

### 4.2.2 Handling multiple attributes

**Computational complexity.** The training loop iterates through all reference images, which implies a linear complexity with respect to the size of the training set. At the same time, it considers all category pairs within an attribute, which means it is quadratic with respect to that. It also iterates through all elements in the joint distribution, so it is exponential with respect to the number of attributes. This can be problematic when we train ITI-Gen on many attributes. For example, in the case of binary attributes with the same number of reference images, the time complexity is $O(Ne^N)$, as Figure 8 depicts.

**Diversity issues in generated images.** Figure 6 illustrates a comparison between ITI-Gen and HPSn. We can observe that ITI-Gen struggles significantly to generate diverse images (the difference between old and young people is almost non-existent, there are many people without eyeglasses, etc.). On the other hand, images generated by HPSn reliably reflect all category combinations.

This behaviour can also be observed in our quantitative results. More specifically, we compute the KL divergence using 104 images in every category combination. The numerical results are displayed in Table 3, on which we can observe that ITI-Gen obtains a KL divergence of approximately 0.3560, while HPSn's is almost equal to 0.

### 4.2.3 Combining ITI-Gen with HPSn

The main motivations for image-guided prompting are handling negations and continuous attributes that are hard to specify with text. As we have shown, negation can be effectively tackled using HPSn, but ITI-Gen

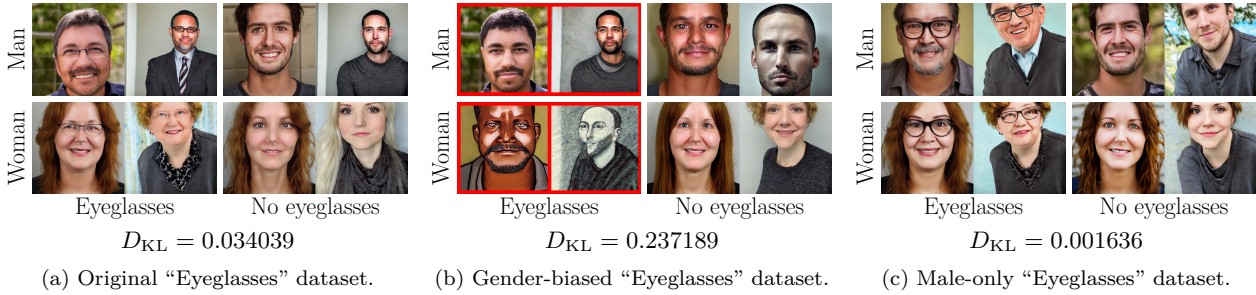

(a) Original "Eyeglasses" dataset.    (b) Gender-biased "Eyeglasses" dataset.    (c) Male-only "Eyeglasses" dataset.

Figure 5: **Ablation study on the diversity of reference datasets.** Generated images for all category combinations of the "Gender" and "Eyeglasses" attributes for all variations of the "Eyeglasses" reference datasets introduced in Table 4 .The complete reference dataset for the "Gender" attribute is used in the three cases. In the three cases, the KL divergence ($D_{\mathrm{KL}}$) is computed using 104 manually labeled samples 4.

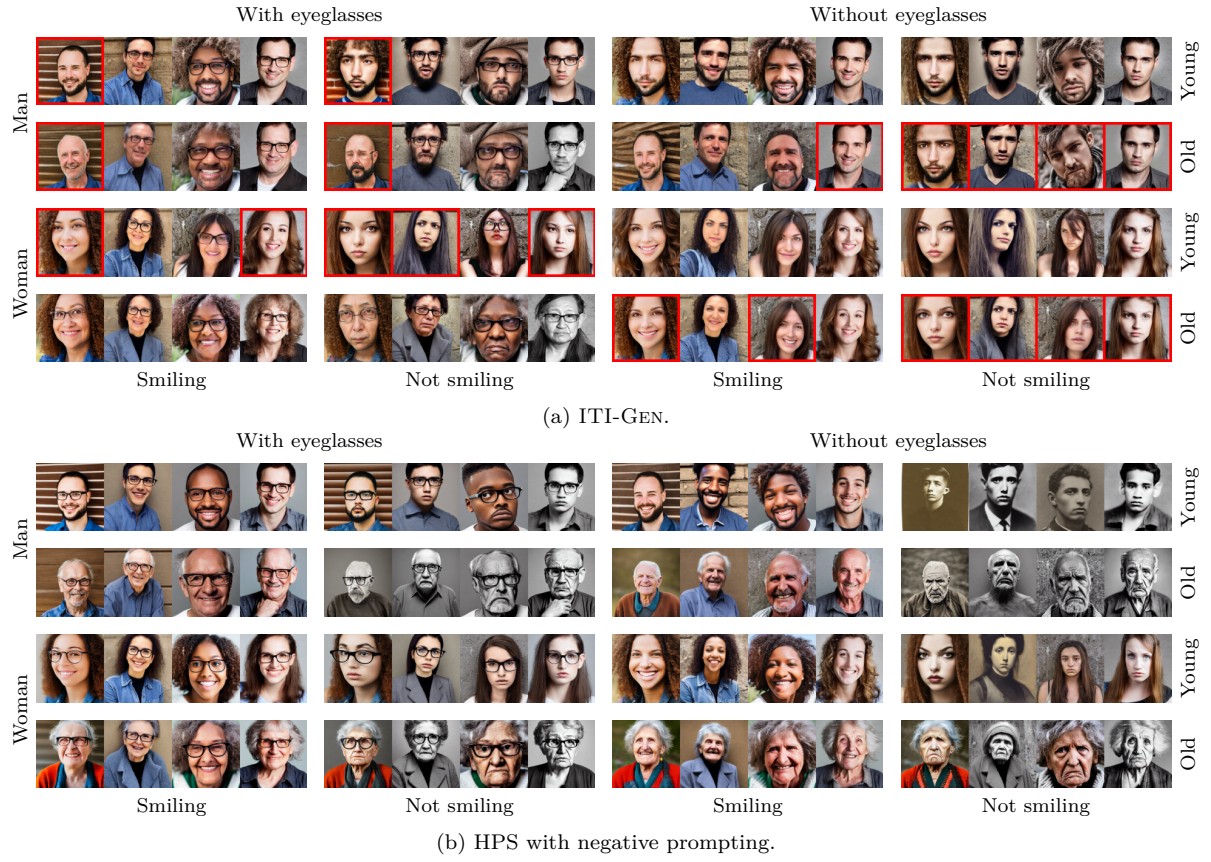

Figure 6: **Qualitative results for multiple attributes.** ITI-GEN and HPSn are compared on generating diverse images with respect to four binary attributes.

is still needed for continuous attributes. A natural question is whether we can combine both methods, and indeed, such integration is feasible.

This can be achieved by appending ITI-GEN's inclusive token after the (positive) prompt in HPSn. In this way, we can just train ITI-GEN on the attributes that are hard to specify with text and leverage the training-free HPSn method for the others. In the experiment described in Figure 7, we get a KL divergence close to zero, and a FID score comparable to ITI-GEN's.

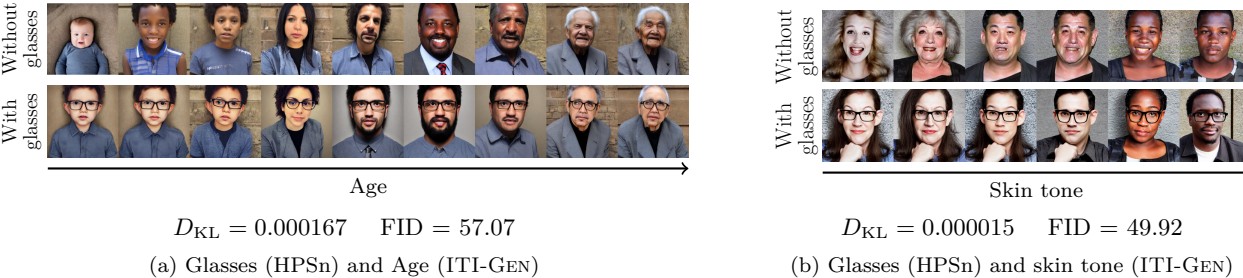

$D_{\mathrm{KL}} = 0.000167 \quad \mathrm{FID} = 57.07$  $D_{\mathrm{KL}} = 0.000015 \quad \mathrm{FID} = 49.92$

(a) Glasses (HPSn) and Age (ITI-GEN)  (b) Glasses (HPSn) and skin tone (ITI-GEN)

Figure 7: **Combining ITI-Gen with HPSn.** ITI-GEN's fair tokens are used for the Age and Skin tone attributes, while the category of the Eyeglasses attribute is hard-prompted in the Stable Diffusion arguments.

## 5 Discussion

In this work, we conduct several experiments to check the validity of the main claims from the original paper. We find that most of them are correct with some small exceptions.

First, we show ITI-GEN (Zhang et al., 2023a) is able to generate diverse high-quality images, according to the low KL divergence and FID scores we obtained (see Tables 1 and 2). We also highlight that it works on multiple domains, such as human faces and landscapes. Moreover, ITI-GEN has plug-and-play capabilities when learned inclusive tokens are applied to similar text prompts (Figure 2). The fair tokens can also be integrated with different text-to-image generators (i.e. Stable Diffusion (Rombach et al., 2022), ControlNet (Zhang et al., 2023b)), as displayed in Figures 1, 2, 3. In addition, we verify that only a few dozen reference images are required, and that training is computationally efficient (Figure 11).

On the other hand, we show that the claim about scalability to multiple attributes does not completely hold. More specifically, our analysis reveals that ITI-GEN needs exponential training time with respect to the number of attributes (Figure 8) and struggles to generate diverse images when we increase the number of attributes (Figure 6). Moreover, we illustrate that ITI-GEN can struggle to disentangle certain attributes (e.g. "Gender" and "Bald" in Figure 4). To solve these issues, we propose HPSn which does not require training and produces more accurate images. At the same time, it seems to handle negations better than ITI-GEN, especially when we use a large number of attributes (Figure 6, Table 1). However, a limitation of HPSn is that it cannot handle attributes that are difficult to specify with text (e.g. skin tone, colorfulness), whereas ITI-GEN excels on these. As we show in Figure 7, ITI-GEN and HPSn can be combined to generate images that are inclusive with respect to multiple attributes, taking advantage of the strong aspects of each method.

### 5.1 What was easy

The paper was well-written and included many examples to make it easier to understand. Additionally, the original code was published on GitHub, with instructions on how to run the training, generation and evaluation scripts.

### 5.2 What was difficult

The main difficulty consisted in understanding the code, since there were some differences with the paper. More specifically, when the directional loss is undefined, it is replaced by a cosine similarity loss between the image and text embeddings.

Moreover, the evaluation script requires a list of classes that is not specified. The authors mention that they had to change the text prompts to tackle the negative prompt issue. They also had to use pre-trained classifiers combined with human evaluations. Thus, it was not possible for us to easily reproduce the KL divergence results.

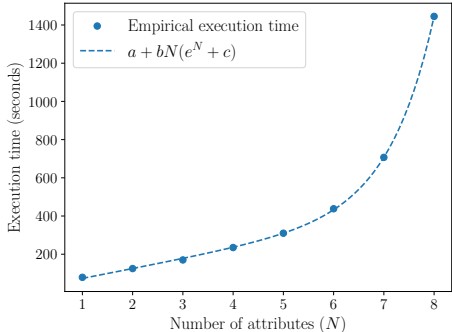

Figure 8: **Computational complexity.** Average training time across 3 executions of ITI-GEN with respect to the number of binary attributes. 400 reference images are used for every attribute, equally distributed between the positive and negative categories.

### 5.3 Communication with original authors

We reached out to the authors via email correspondence regarding the quality of the generated images and the missing code for combining ITI-GEN with ControlNet. They replied quickly and cleared up the discrepancies in our results. However, we did not receive from the authors any additional code for integrating ITI-GEN with ControlNet, because they were still working on it. They assured us they would upload the code on their original GitHub repository as soon as possible, so that we can cross-verify our implementations.

## 6 Ethical and Social Impact

As ITI-GEN is guided visually, we need a set of reference images. This raises some concerns regarding inclusiveness and privacy.

Even though the output images are usually inclusive with respect to the attributes of interest, they might not be with respect to others. This can be problematic, because someone aiming to increase diversity could actually be introducing other biases without realizing it. Thus, it is important to ensure that the reference images are not biased, and conduct some sort of quality evaluation on the outputs.

Another important aspect is data protection. If ITI-GEN is trained on a publicly available dataset, we need to have permission to do so. Inspired by Carlini et al. (2023), we wonder if an attacker could recover the training images using the inclusive tokens, which would be problematic if ITI-GEN was trained on a private dataset.

All in all, we agree with the original authors' that carefully considering potential risks of ITI-GEN is essential to avoid possible negative consequences.

### Acknowledgments

We would like to thank Carlo Bretti from the University of Amsterdam for his useful feedback and guidance during the process of writing the paper.

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

## A    KL Divergences on the CelebA dataset

Table 5 shows the performance on diversity of ITI-GEN and the baseline methods on binary attributes.

Table 5: **Inclusiveness with respect to single attributes.** KL Divergence for all attributes from CelebA dataset. We include results for ITI-GEN trained for 10, 20 and 30 epochs. The generated images (208 per attribute) were classified using CLIP, and, for some attributes, also manually (last column).

| Attribute | SD | HPS | HPSn | ITI-GEN | | | |
| --- | --- | --- | --- | --- | --- | --- | --- |
| | | | | 10 epochs | 20 epochs | 30 epochs | 30 (man.) |
| 5 o'Clock Shadow | 0.007833 | 0.076791 | 0.016782 | **0.000185** | 0.007833 | 0.008748 | - |
| Arched Eyebrows | 0.355570 | 0.120493 | **0.011881** | 0.101287 | 0.072061 | 0.083041 | - |
| Attractive | 0.072061 | **0.001156** | 0.036694 | 0.158797 | 0.130812 | 0.203595 | - |
| Bags Under Eyes | 0.325839 | 0.125592 | **0.001156** | 0.255716 | 0.120493 | 0.034089 | - |
| Bald | 0.530124 | 0.048114 | **0.000000** | 0.042202 | 0.068324 | 0.101287 | 0.000178 |
| Bangs | 0.263824 | 0.263824 | **0.000000** | 0.000416 | 0.002962 | 0.003749 | - |
| Big Lips | 0.158797 | 0.064693 | 0.064693 | 0.001156 | **0.000416** | 0.002267 | 0.008915 |
| Big Nose | 0.355570 | **0.013420** | **0.013420** | 0.280577 | 0.316377 | 0.272110 | 0.201355 |
| Black Hair | 0.316377 | 0.110655 | **0.002962** | 0.196770 | 0.263824 | 0.232416 | 0.020136 |
| Blond Hair | 0.638921 | 0.020527 | **0.001665** | 0.007833 | 0.006672 | 0.007833 | - |
| Blurry | **0.016782** | 0.217696 | 0.022544 | 0.036694 | 0.020527 | 0.031584 | - |
| Brown Hair | 0.421958 | 0.158797 | **0.003749** | 0.272110 | 0.224977 | 0.217696 | - |
| Bushy Eyebrows | 0.105913 | 0.376598 | **0.000046** | 0.002962 | 0.016782 | 0.011881 | - |
| Chubby | 0.298080 | 0.514949 | **0.000416** | 0.115515 | 0.164783 | 0.183554 | - |
| Double Chin | 0.272110 | 0.446529 | 0.002267 | 0.031584 | 0.026869 | **0.001665** | - |
| Eyeglasses | 0.398692 | 0.387506 | **0.000000** | 0.016782 | 0.020527 | 0.015053 | 0.000000 |
| Goatee | 0.345427 | 0.064693 | **0.010438** | 0.170903 | 0.158797 | 0.170903 | - |
| Gray Hair | 0.693147 | 0.365957 | **0.000046** | 0.079859 | 0.088094 | 0.092379 | 0.035989 |
| Heavy Makeup | 0.434070 | 0.136156 | **0.000185** | 0.096776 | 0.064693 | 0.045107 | 0.113232 |
| High Cheekbones | 0.472577 | **0.003749** | 0.096776 | 0.434070 | 0.335520 | 0.355570 | - |
| Male | 0.000740 | **0.000000** | **0.000000** | **0.000000** | **0.000000** | **0.000000** | 0.000000 |
| Mouth Slightly Open | 0.083921 | 0.110655 | 0.110655 | 0.007833 | **0.000046** | 0.000185 | - |
| Mustache | 0.545925 | 0.158797 | **0.000000** | 0.280577 | 0.280577 | 0.280577 | 0.000000 |
| Narrow Eyes | 0.136156 | 0.051223 | 0.026869 | 0.026869 | **0.000000** | 0.006672 | 0.000000 |
| No Beard | 0.232416 | 0.500334 | 0.693147 | **0.013420** | 0.031584 | 0.042202 | 0.000000 |
| Oval Face | 0.376598 | **0.010438** | 0.170903 | 0.240016 | 0.101287 | 0.170903 | - |
| Pale Skin | 0.196770 | **0.000416** | **0.000416** | 0.004630 | 0.006672 | 0.001156 | 0.000185 |
| Pointy Nose | 0.500334 | **0.061169** | 0.164783 | 0.545925 | 0.562439 | 0.500334 | - |
| Receding Hairline | 0.410171 | 0.514949 | **0.000740** | 0.136156 | 0.141624 | 0.141624 | |
| Rosy Cheeks | 0.486224 | 0.051223 | **0.024658** | 0.345427 | 0.345427 | 0.421958 | - |
| Sideburns | 0.514949 | **0.026869** | 0.034089 | 0.387506 | 0.410171 | 0.562439 | - |
| Smiling | 0.280577 | 0.000046 | 0.000046 | **0.000000** | **0.000000** | **0.000000** | |
| Straight Hair | 0.057750 | 0.662690 | **0.000000** | 0.002267 | 0.000185 | 0.002267 | - |
| Wavy Hair | 0.446529 | 0.240016 | **0.000185** | 0.115515 | 0.072061 | 0.064693 | - |
| Wearing Earrings | 0.355570 | 0.472577 | **0.000416** | 0.398692 | 0.446529 | 0.421958 | - |
| Wearing Hat | 0.662690 | 0.545925 | **0.000046** | 0.042202 | 0.136156 | 0.141624 | - |
| Wearing Lipstick | 0.579782 | 0.500334 | **0.000046** | 0.247782 | 0.307126 | 0.325839 | - |
| Wearing Necklace | 0.662690 | 0.307126 | **0.001156** | 0.355570 | 0.355570 | 0.325839 | - |
| Wearing Necktie | 0.514949 | 0.545925 | **0.001156** | 0.009088 | 0.001665 | 0.002267 | - |
| Young | 0.693147 | **0.000000** | **0.000000** | 0.000046 | 0.001665 | 0.026869 | 0.001156 |

## B    Compatibility with ControlNet

We present additional results on the compatibility of ITI-GEN with ControlNet (Zhang et al., 2023b) in Figure 9.

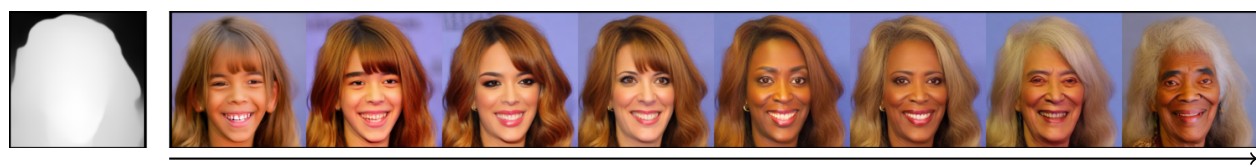

Age

(a) Condition: depth map. Prompt: "a headshot of a female".

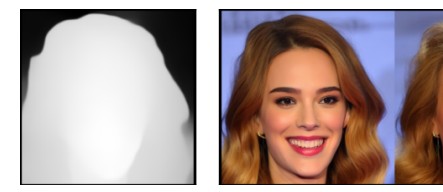 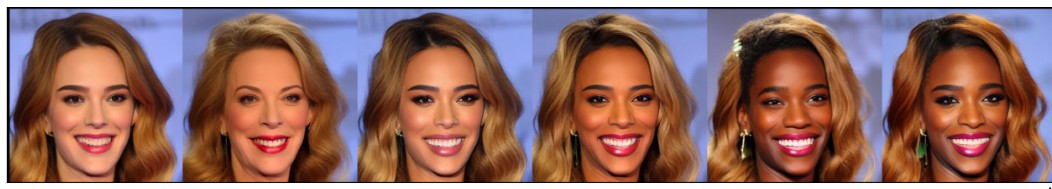

Skin tone

(b) Condition: depth map. Prompt: "photo of a famous woman".

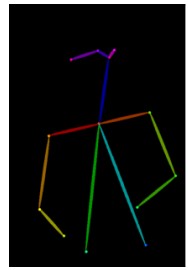 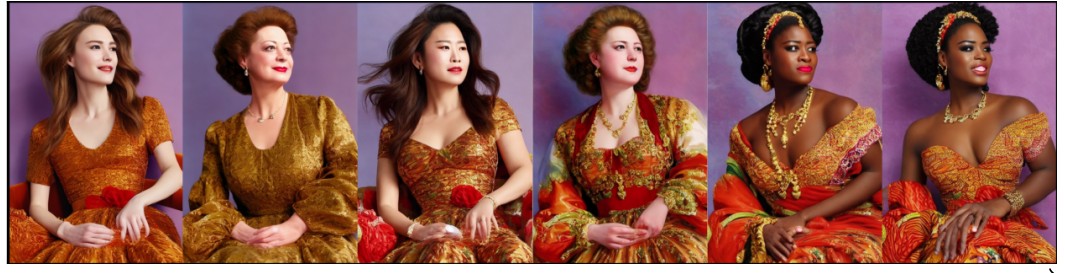

Skin tone

(c) Condition: human pose map. Prompt: "photo of a famous woman".

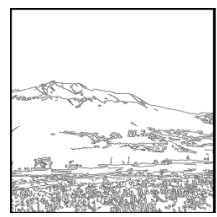 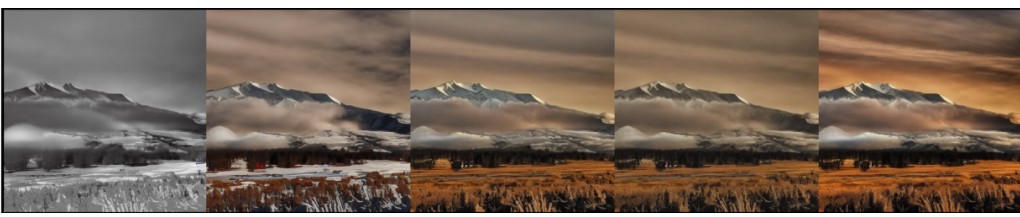

Colorfulness

(d) Condition: canny edge map. Prompt: "photograph of mount katahdin".

Figure 9: **Compatibility with ControlNet.** ITI-GEN is able to improve diversity in images generated with ControlNet with respect to the attributes of interest given different conditions. Human images (a), (b), (c) are generated using inclusive tokens trained on "a headshot of a person", while scene images (d) are generated using inclusive tokens trained on "a natural scene".

## C Proxy features

We provide additional examples of proxy features used by ITI-GEN in Figure 10.

## D Low data requirements

Figure 11 illustrates how ITI-GEN is able to improve diversity using only a few reference images.

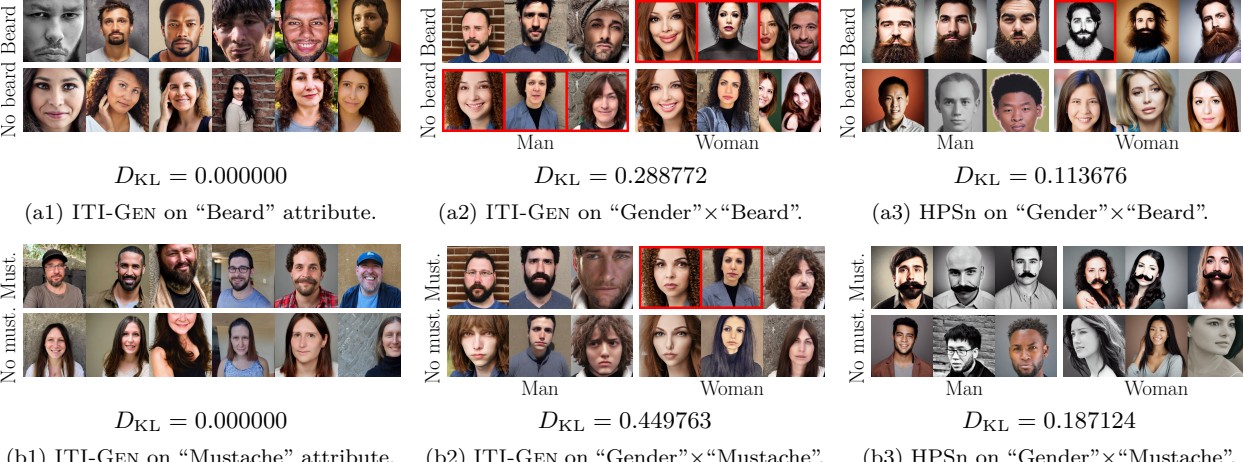

$D_{\mathrm{KL}} = 0.000000$

(a1) ITI-GEN on "Beard" attribute.

$D_{\mathrm{KL}} = 0.288772$

(a2) ITI-GEN on "Gender"×"Beard".

$D_{\mathrm{KL}} = 0.113676$

(a3) HPSn on "Gender"×"Beard".

$D_{\mathrm{KL}} = 0.000000$

(b1) ITI-GEN on "Mustache" attribute.

$D_{\mathrm{KL}} = 0.449763$

(b2) ITI-GEN on "Gender"×"Mustache".

$D_{\mathrm{KL}} = 0.187124$

(b3) HPSn on "Gender"×"Mustache".

Figure 10: **Proxy features are used by the model for certain attributes.** In (a1) and (b1) we see how ITI-GEN seems to be using "Gender" as a proxy for "Beard" and "Mustache". In (a2) and (b2) we see how ITI-GEN fails to disentangle the attributes. In (a3) and (b3) we see HPSn's results. It struggles sometimes to generate women with beard, but works well with the combination of "Gender" and "Mustache". In (b) and (c), the KL Divergence is computed using 104 manually labeled samples.

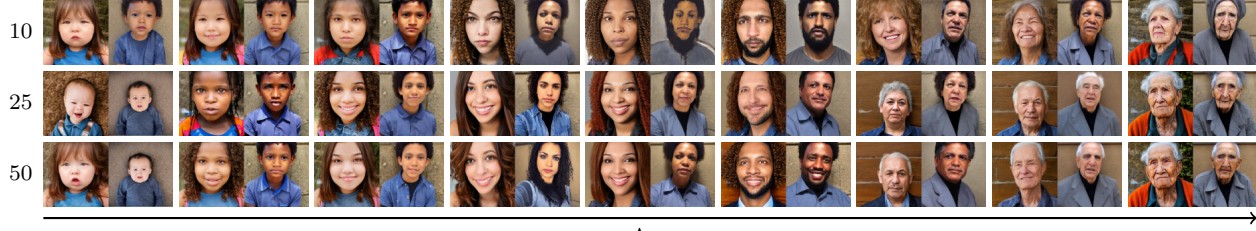

(a) Prompt: "a headshot of a person".

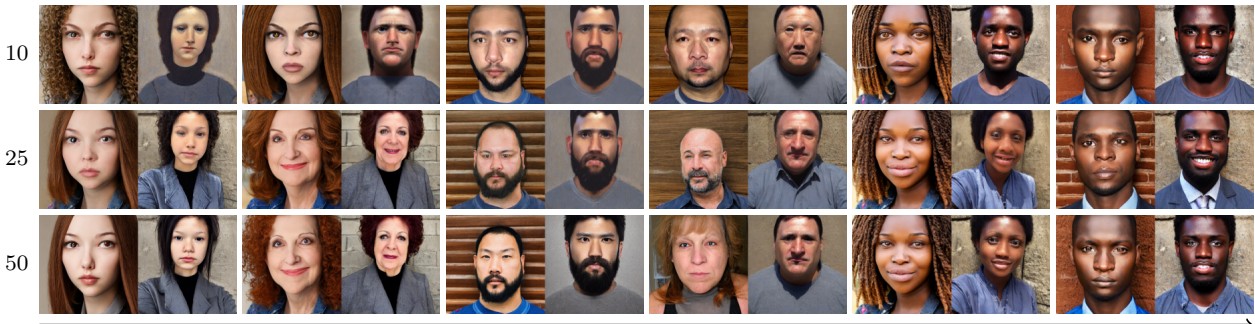

(b) Prompt: "a headshot of a person".

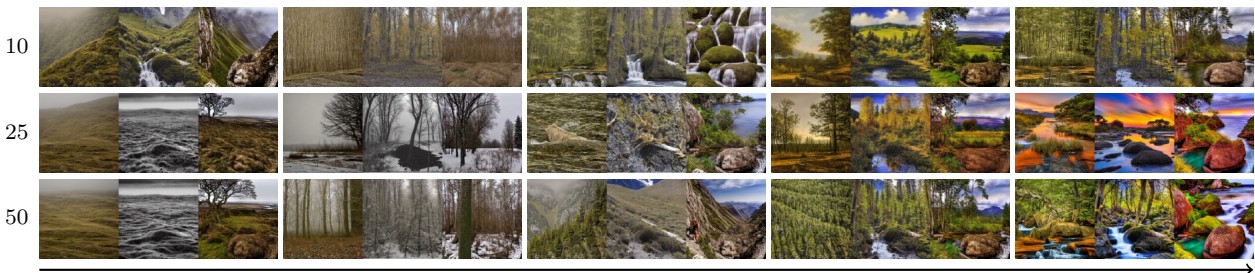

(c) Prompt: "a natural scene".

Figure 11: **Effect of of the size of the reference dataset.** ITI-GEN is able to improve diversity with respect to attributes such as "Age", "Skin tone" and "Colorfulness" using only a few reference images (10, 25 and 50).

