# OpenReview forum: "Reproducibility Study of "ITI-GEN: Inclusive Text-to-Image Generation""
_TMLR — Accepted by TMLR_

### Review · Reviewer_RH1q · 2024-03-04

**Summary Of Contributions:**

This paper attempts to replicate and evaluate the findings of "ITI-Gen: Inclusive Text-to-Image Generation" by Zhang et al. (2023a).

The paper confirms several of the original authors' claims about ITI-Gen, including (1) its ability to enhance the diversity and quality of generated images, (2) its scalability across different domains, (3) its plug-and-play capabilities, and (4) its computational efficiency.

However, the paper also identifies some limitations of ITI-Gen. It occasionally relies on undesirable attributes as proxy features and struggles to disentangle certain correlated attributes, such as gender and baldness. Additionally, as the number of attributes considered increases, the training time rises exponentially, and ITI-Gen faces challenges in generating inclusive images for all elements in the joint distribution.

To address these issues, the paper proposes the use of Hard Prompt Search with negative prompting. This method does not require training and is more effective at handling negation than the standard Hard Prompt Search.

**Audience:**

Yes

**Broader Impact Concerns:**

Given that the generated images involve human faces, it is recommended to incorporate a Broader Impact Statement or an ethical statement in the submission.

**Claims And Evidence:**

Yes

**Requested Changes:**

As mentioned in the comments above,

(1) The section describing the model could be elaborated upon. It should reflect the authors' understanding rather than being a mere paraphrase of the original paper.
(2) The paper could benefit from including more details on the reproduction process.
(3) The authors should clarify the computational and time complexities associated with using HPSn.
(4) While HPSn can handle attributes like "Gender" and "Bald" better than ITI-Gen, it struggles with attributes difficult to define textually, such as skin tone. Could the authors propose a solution that encompasses a broader range of attributes? This could be a significant new contribution beyond merely evaluating existing work.
(5) Besides FID, have the authors considered conducting human evaluations to assess the quality of the generated images?
(6) Regarding communication with the original authors of ITI-Gen, they seem cooperative. Although the requested additional code might not be available at the time of this submission, the authors could cross-verify their implementation with the ITI-Gen authors since they have developed the code.
(7) In addition to the issues identified in the paper, such as ITI-Gen's struggle to disentangle attributes like "Gender" and "Bald," are there aspects of the ITI-Gen model that the authors believe could be improved? For instance, the authors mention problems when training ITI-Gen on numerous attributes; have any potential solutions or experiments been attempted to address this?
(8) It would be beneficial to include a discussion on the experimental results, specifically addressing which results align with the claims in the ITI-Gen paper and which do not, along with potential explanations for any discrepancies.
(9) For the authors' implementation, such as ControlNet, adding implementation details to the submission would be valuable. The reproduction work should be as comprehensive as possible.

**Strengths And Weaknesses:**

Strengths:

(1)  The paper successfully replicates the results from the original study, validating the claims presented in Section 2 by reproducing select experiments from Zhang et al. (2023a).
(2) It thoroughly examines all contributions made by Zhang et al. (2023a), including aspects like inclusive and high-quality generation, scalability across different domains, plug-and-play capabilities, data and computational efficiency, and scalability to multiple attributes. The reproducibility scope is clearly defined and easy to follow.
(3) Motivated by the attributes where ITI-Gen underperforms, the authors conduct experiments to analyze the impact of diversity and entanglement in the reference image datasets.
(4) Beyond mere replication, the paper explores the addition of negative prompts to Stable Diffusion as an alternative approach to handling negations in natural language, comparing the outcomes with ITI-Gen.
(5) The code used for reproduction is made available. The authors have implemented and shared additional code for integrating ITI-Gen with other models like ControlNet.


Weaknesses:

(1) The replication of an existing study lacks a comprehensive introduction to the model in the evaluated paper. Specifically, Section 3.1 provides an insufficient description of the model. It would be beneficial to include a detailed explanation of the model architecture, the functionality of each component, the loss function, the unique aspects of the ITI-GEN proposal, and how these features address existing issues. A thorough introduction and clarification would offer readers a clearer understanding of the research and ensure that the authors have accurately interpreted the evaluated paper.
(2) Reproducibility research aims to validate the claims regarding model correctness, code accuracy, and experimental parameters made in the published paper. Therefore, it is recommended to incorporate as many replication details as possible. For instance, the authors mention, "The authors do not discuss the hyperparameters used for image generation. We found the scale hyperparameter $\lambda$ to be the most important one after some experiments." It would be advisable to provide specifics on how the authors determined the importance of $\lambda$ and the experiments conducted to reach this conclusion.

---

> ### Author Response · Authors · 2024-04-04
>
> Dear reviewer,
>
> Thank you for your detailed feedback. We have uploaded a new version of the paper. Below we specify the modifications we made to address the changes you requested:
>
> 1.  Indeed, the description of the model could have been elaborated more. We added more details on ITI-GEN in Section 3.1, especially regarding the replacement of the directional alignment loss by a similarity loss, which is not specified in the original paper (but rather in the provided code).
>
> 2. Regarding adding more information about the reproducibility process, we now include more information about integrating ITI-GEN with ControlNet in Section 3.4, since this code is not provided by the original authors.
> Regarding Weakness 2 (the influence of λ on image generation), we have decided to remove the explanation from Section 3.3, since at the end we just used the default hyperparameters of the different repositories.
>
> 3. Regarding HPSn, we mention in the Discussion that HPSn does not require training. Therefore, the only cost is the one associated with the generation of images, which we specify in Section 3.5.
>
> 4. In Section 4.2.3 of  the new version of the paper, we also show that HPSn can be combined with ITI-GEN to solve the major limitations of ITI-GEN (long training times and inclusiveness issues with multiple attributes) and HPSn (inability to handle attributes that are hard to specify with text).
>
> 5. We have performed human evaluations to compare ITI-GEN with HPS and HPSn, which is explained in Section 3.4. We show the results and discuss them in Section 4.1.1 of the new version of the paper.
>
> 6. We discussed with the authors and they assured us they would upload the code in the following weeks. Therefore, we will cross-verify the code with them as soon as they make it public. We have updated this information in Section 5.3, where we talk about the communication with the authors.
>
> 7. We point out two issues with ITI-GEN: the training time and the inclusiveness with multiple attributes. The training time problem cannot be tackled easily, as one would have to change the training procedure completely. However, we propose using HPSn (that does not have any of these two issues) in Section 4.2.2 and combining HPSn with ITI-GEN in Section 4.2.3 of the new version.
>
> 8. We have expanded the discussion section (Section 5) to make more explicit which claims hold and which ones do not.
>
> 9. This is a valuable point. We added more insights related to our implementation in Section 3.4.
>
> We also added a section at the end called “Ethical and Social Impact” discussing these issues.
>
> Thank you once again for the suggestions.

---

### Review · Reviewer_Fur8 · 2024-03-20

**Summary Of Contributions:**

This paper aims to reproduce "Inclusive Text-to-Image Generation", which presents ITI-Gen to improve inclusiveness in text-to-image (T2I). From the experiments, most of the claims hold, including the improvement in quality as well as diversity, the scalability to different domains, and its plug-and-play capability. They also highlight the issue of undesired proxy features, poor disentanglement, and excessive training time costs, which are not covered in the original paper. Beyond this, they leverage hard prompt search with negative prompts, which may handle potential negation and address these issues.

**Audience:**

Yes

**Claims And Evidence:**

Yes

**Requested Changes:**

I think this version is in general pretty good. I propose some points that can make this draft more robust in weakness.

**Strengths And Weaknesses:**

**Strength**
+ This paper is well-written and easy to follow.
+ They successfully reproduce the training, evaluation, and results. Apart from that, they also point out some potential issues, which are valuable for this research direction.
+ They provide comprehensive quantitative analysis as well as qualitative comparison.
+ They have open-sourced their reproduced code base.

**Weakness**
+ As they mention the training cost, the paper should contain learning curves for each dataset to indicate this issue. Also, the exact learning record (both training loss and validation performance) is crucial and helpful for this kind of rerpoduce report.
+ The integration with ControlNet seems not that well. For example, in Fig. 3&8a, the girls change too much rather than just age. Furthermore, different attributes should be considered and demonstrated to support this integration.
+ Since this tackles a real-world application/issue, it will be better to have a human evaluation using the reproduced results to investigate human preference in various aspects.

---

> ### Author Response · Authors · 2024-04-01
>
> Dear reviewer,
>
> Thank you for your feedback. We are finishing an updated version of the paper based on your suggestions. Before submitting it, we would like to add some clarifications related to the first two points from the weakness section:
>
> - Although it is possible to plot the evolution of the loss(es) during training, we don’t think this has relation with the training cost, as it will be exponential with respect to the number of attributes regardless of the number of epochs, which has a linear impact.
> - Regarding ControlNet, the authors mention in the original paper that they _“observe an intriguing feature where the newly introduced tokens may implicitly entangle other biases or constasts inherent in the reference image sets, such as clothing style”_. They also mentioned that _“disentanglement of attributes is not the primary concern of this study”_, which we agree with. We believe that the integration with ControlNet holds for the attributes of interest (e.g., Age in our case), even though the images contain other biases related to clothing, race or hair color which can be caused by the lack of diversity of the reference images during training. However, we do think it is important for the reader to know this, so we will add this information in the captions of the figures. We will also add more examples for other attributes (e.g. Skin tone).
>
> We hope this clarifies some of the concerns you expressed in your review. Please let us know if you have further insights on these aspects.
>
> We will come back to you with the new version of the paper in 2-3 days.

---

> ### Author Response · Authors · 2024-04-04
>
> Dear reviewer:
>
> Thank you again for your feedback. We have uploaded a new version of the paper. Below we specify the modifications we made to address the changes you requested:
>
> - We specify the scope and objectives of the experiments with ControlNet in section 4.1.3.
> - We understand that we should test the integration on more attributes, especially for face attributes. Thus, we also tested the integration for Skin color using pose and depth as additional feature maps. We have included the results for this new experiment in Appendix B.
> - We have performed human evaluations to compare ITI-GEN with HPS and HPSn. We show the results and discuss them in section 4.1.1 of the new version of the paper.

---

### Review · Reviewer_j9Wj · 2024-03-22

**Summary Of Contributions:**

This paper conducts a reproducibility study on another paper: “ITI-GEN: Inclusive Text-to-Image Generation", it also has the following contributions:
- Investigated the impact of diversity and entanglement in reference image datasets, particularly where ITI-Gen struggled.
- Explored the use of negative prompts in image generation, comparing it with existing methods.
- Enhanced model performance by fixing bugs, integrated with other models, and provided code for easy replication of experiments.

**Audience:**

No

**Claims And Evidence:**

Yes

**Requested Changes:**

The authors may need to convince us the novelty and importance of this reproductivity study

**Strengths And Weaknesses:**

Strengths:
- This paper conducts a lot of experiments related to the reproduction of “ITI-GEN: Inclusive Text-to-Image Generation”
- The paper is clear with figures and tables
- The paper proposes to use Hard Prompt Search with negative prompting, which does not require training and is more effective at handling negation than the standard Hard Prompt Search.


Weakness:
- The paper on reproducibility study lacks novelty as it primarily focuses on replicating experiments from a previous work without introducing any new methods, insights, or advancements to the field. As such, its contribution to the scientific community is limited, and it does not significantly add to the existing body of knowledge.
- While I recognize the importance of reproducibility studies in validating previous research findings, the novelty of such studies lies in providing additional insights or uncovering discrepancies that significantly impact the original findings or methodology. Unfortunately, upon reviewing your manuscript, it appears that the findings do not substantially contribute to the existing body of knowledge or challenge previous results in a meaningful way.
- Although the study is well-executed and adheres to rigorous methodology, it lacks innovation, new perspective, more insights, interpretations, or advancements that enhance our understanding of the subject matter.

---

> ### Author Response · Authors · 2024-04-01
>
> Dear reviewer,
>
> Thank you for your feedback. After reflecting on it, we would like to make the following remarks:
>
> - This is a reproducibility study carried out under the guidelines of the MLRC (https://reproml.org/), which is partnering with TMLR for the first time. Thus, our first aim was to verify the main claims of the original paper.
> - We point out two drawbacks of the original paper: (1) the usage of proxy features and (2) issues when handling multiple attributes. Regarding the former, we discovered that ITI-GEN is unable to disentangle pairs of (correlated) attributes, which results in non-inclusive generation. Regarding the latter, we showed that training time grows exponentially with the number of features, and that diversity diminishes significantly.
> - In addition, we propose using HPSn, a method that handles negation more effectively than HPS. In the new version of the paper that we will submit in the upcoming days, we will also provide results on how HPSn can be combined with ITI-GEN to tackle both method’s limitations.
>
> We hope this clarifies the scope of our study and our additions to the original work. We will come back to you with the new version of the paper in 2-3 days.

---

> ### Author Response · Authors · 2024-04-04
>
> Dear reviewer:
>
> Thank you again for your feedback. We have uploaded a new version of the paper. In addition to what we mentioned in our previous comment,  we would like to point out that we have added a new section (4.2.3.) showing the potential of combining ITI-GEN with HPSn with the intention to add value to our research.

---

### Public Comment · ~Christopher_T.H_Teo1 · 2024-07-26
**Questions on ITI-Gen Quality**

Dear Authors,

I enjoyed reading your paper on reproducing ITI-Gen.
However, in our experiments, we do observe some degraded samples by ITI-Gen (10 samples for your reference with attribute High Cheekbones): https://drive.google.com/drive/folders/1MbOXSTUUSS9f3EO-wfTQ9IA08AODPR2U?usp=share_link

In this experiment, we follow the same setting in ITI-Gen with the base prompt  “A headshot of a person” to generate 500 samples per attribute. Are these observations consistent with your quality assessment?
Thank you.

---

### Decision · Action_Editor_JMjF · 2024-05-19

**Recommendation:** Accept as is

**Comment:**

All reviewers agree with the contribution in reproducing the results. The concerns about the novelty are not issues of TMLR.

**Audience:**

It would be interesting to TMLR audience.

**Claims And Evidence:**

This paper aims to reproduce the results of "Inclusive Text-to-Image Generation".  It does a reasonable job though it also lacks new insight and discoveries.